# NEURAL PARAMETER REGRESSION FOR EXPLICIT REPRESENTATIONS OF PDE SOLUTION OPERATORS

**Konrad Mundinger**[*1,2]**, Max Zimmer**[1] **& Sebastian Pokutta**[1,2]
[1]Department for AI in Society, Science, and Technology, Zuse Institute Berlin, Germany
[2]Institute of Mathematics, Technische Universität Berlin, Germany
{mundinger, zimmer, pokutta}@zib.de

## ABSTRACT

We introduce *Neural Parameter Regression (NPR)*, a novel framework specifically developed for learning solution operators in Partial Differential Equations (PDEs). Tailored for operator learning, this approach surpasses traditional DeepONets (Lu et al., 2021a) by employing Physics-Informed Neural Network (PINN, Raissi et al., 2019) techniques to regress Neural Network (NN) parameters. By parametrizing each solution based on specific initial conditions, it effectively approximates a mapping between function spaces. Our method enhances parameter efficiency by incorporating low-rank matrices, thereby boosting computational efficiency and scalability. The framework shows remarkable adaptability to new initial and boundary conditions, allowing for rapid fine-tuning and inference, even in cases of out-of-distribution examples.

## 1 INTRODUCTION

Partial Differential Equations (PDEs) are central to modeling complex physical phenomena across diverse application areas. Traditional approaches often rely on numerical methods due to the scarcity of closed-form solutions for most PDEs. The emergence of Physics-Informed Neural Networks (PINNs) (Raissi et al., 2019) has revolutionized this domain. PINNs integrate the governing PDEs, along with initial and boundary conditions, into the loss function of a Neural Network (NN), enabling self-supervised learning of approximate PDE solutions.

While PINNs are mostly employed to solve single instances of (possibly parametric) PDEs, Physics-Informed DeepONets (Wang et al., 2021b) have extended their capabilities to the learning of solution operators, i.e., mappings from initial conditions to solutions. We introduce *Neural Parameter Regression (NPR)*, advancing operator learning by combining Hypernetworks (Ha et al., 2017) with Physics-Informed operator learning. Our approach significantly deviates from existing methods (de Avila Belbute-Peres et al., 2021; Zanardi et al., 2023; Cho et al., 2023) by parametrizing the output network as a low-rank model which by design satisfies the initial condition and is not required to learn an identity mapping. We refer to Section A.1 for a detailed discussion of related work.

Our main contribution is a novel combination of Hypernetwork approaches with PINN techniques in the setting of operator learning. We experimentally demonstrate the ability of our approach to accurately capture (non-)linear dynamics across a wide range of initial conditions. Additionally, we showcase the remarkably efficient adaptability of our model to out-of-distribution examples. Section 2 introduces related concepts and our proposed method, Section 3 describes the training procedure and presents experimental results, and Section 4 concludes the paper. Detailed discussions and extended results can be found in the appendix.

## 2 METHODS

*PINNs* have emerged as a powerful tool for approximating solutions of PDEs by employing automatic differentiation to construct a loss function which is zero if and only if the NN solves the PDE.

---

*Corresponding author.

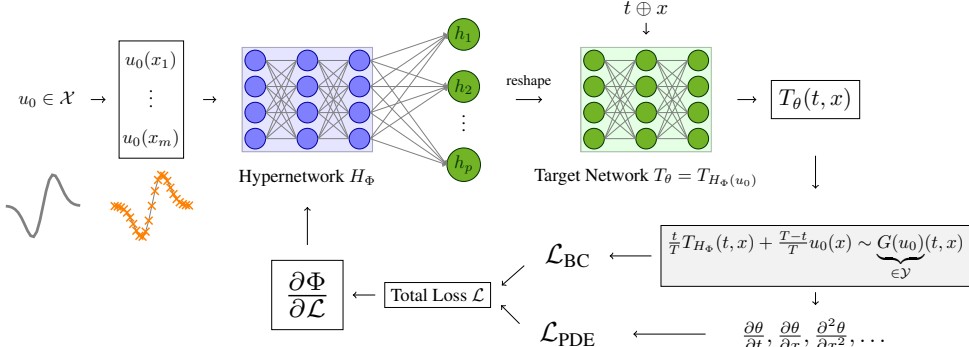

Figure 1: The proposed architecture: The Hypernetwork $H_\Phi$ maps an initial condition to another function parametrized by $T_{H_\Phi}$, hence approximating a mapping between function spaces.

The *DeepONet* framework (Lu et al., 2021a) employs NNs to approximate operators in Banach spaces. It evaluates an operator $G$ through $G(u)(y)$ for functions $u$ and vectors $y$, using a Branch Network $G_{\text{branch}}$ for encodings of $u$ and a Trunk Network $G_{\text{trunk}}$ for encodings of $y$, combined via the dot product. Initially designed for supervised operator learning, its extension to self-supervised learning of PDE solution operators is achieved through *Physics-Informed DeepONets* (Wang et al., 2021b). On the other hand, *Hypernetworks* (Ha et al., 2017) are a meta-learning approach, where one NN (the Hypernetwork) learns the parameters of another NN (the Target Network). We refer the reader to the appendix for more in-depth descriptions of these techniques.

We consider initial boundary value problems (IBVPs) of the form

$$\partial_t u(t,x) = \mathcal{N}(u(t,x)) \qquad \text{for } (t,x) \in [0,T] \times \Omega \tag{1a}$$
$$u(0,x) = u_0(x) \qquad \text{for } x \in \Omega \tag{1b}$$
$$\mathcal{B}(u(t,x)) = 0 \qquad \text{for } (t,x) \in [0,T] \times \partial\Omega. \tag{1c}$$

Here, $\Omega \subset \mathbb{R}^d$ for some $d \in \mathbb{N}$, $T > 0$ is the final time, $u_0$ the initial condition, $\mathcal{B}$ describes the boundary condition, and $\mathcal{N}$ is an operator typically composed of differential operators and forcing terms. Precisely, we are interested in approximating the *Solution Operator*, i.e., the mapping

$$G : \mathcal{X} \supset K \to \mathcal{Y}, \; u_0 \mapsto \big((t,x) \mapsto u(t,x)\big), \tag{2}$$

between the infinite-dimensional Banach spaces $\mathcal{X}$ and $\mathcal{Y}$, where $K$ is compact. In the following we assume that Equation 1 is well-posed, i.e., that the solution (in a suitable sense) exists and is unique.

We propose *(Physics-Informed) Neural Parameter Regression (NPR)*, a novel architecture for learning solution operators of PDEs. The architecture is depicted in Figure 1. It consists of a *Hypernetwork* $H_\Phi$ with parameters $\Phi$, which takes the role of the Branch Network in the DeepONet. By mapping a discretization of a function $u_0$ to a $p$-dimensional vector, we compute the parameters $\theta$ of another NN denoted by $T_\theta$. Further, we pass $(t,x)$ through this learned network and obtain an approximation of $G(u_0)(t,x)$, such that $H_\Phi(u_0)$ explicitly approximates the function $G(u_0)$.

A solution operator of an IBVP faces the challenge of implicitly learning an identity mapping, as $G(u_0)(0,\cdot) = u_0$ must hold for all $u_0$. Since learning the identity is a notoriously hard task (Hardt & Ma, 2017), an essential part of our approach is to enforce the initial conditions in the Target Networks by parametrizing the output as the *deviation from the initial condition over time*. This approach is known as *Physics-Constrained NNs* (Lu et al., 2021b) and we refer to Section A.6 for a detailed description. Precisely, we reparametrize the output of $T_\theta$ as

$$\hat{T}_{H_\Phi(u_0)}(t,x) = \frac{t}{T} T_{H_\Phi(u_0)}(t,x) + \frac{T-t}{T} u_0(x). \tag{3}$$

Conceptually, the network $H_\Phi$ approximates a mapping between the function spaces $\mathcal{X}$ and $\mathcal{Y}$. Effectively however, it is a mapping from $\mathbb{R}^{d_{\text{enc}}}$ to $\mathbb{R}^p$, where $d_{\text{enc}}$ is the encoding dimension (number of sensors in Lu et al. (2021a)). Since the input and output dimensions are low, the majority of parameters is in the hidden weights, which scale in $\mathcal{O}(n_{\text{hidden}} d_{\text{hidden}}^2)$, where $n_{\text{hidden}}$ is the number

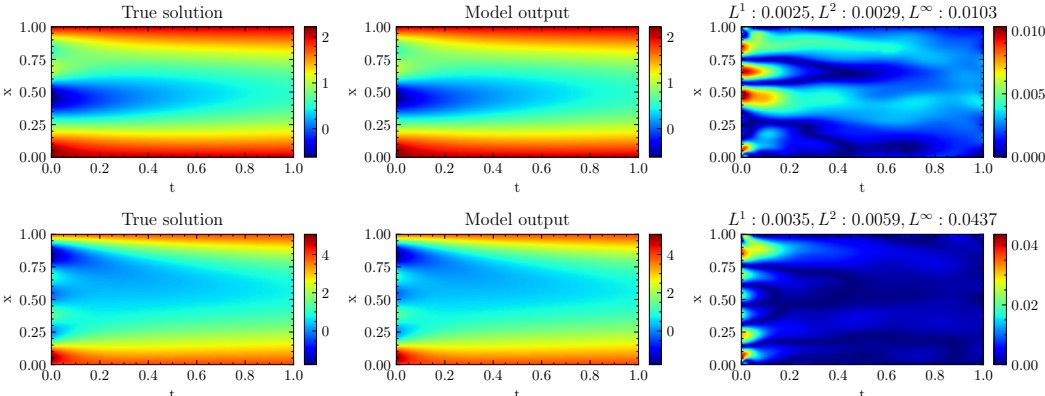

Figure 2: The heat equation: The first and second row correspond to $u_0(x) = 0.5\sin(4\pi x) + \cos(2\pi x) + 0.3\cos(6\pi x) + 0.8$ and the $u_0(x) = \sum_{n=1}^{3}(\sin(n\pi x) + \cos(n\pi x)) + 1$, respectively. The columns shows the reference solutions, the results of NPR and the absolute differences, respectively.

of hidden layers and $d_{\text{hidden}}$ is the hidden dimension. By parametrizing the hidden weights as a product of two matrices $A_i$ and $B_i$ of maximal rank $r$ (Hu et al., 2022), we can reduce the number of parameters to $\mathcal{O}(n_{\text{hidden}} d_{\text{hidden}} r)$. More details can be found in Section A.8.

## 3 EXPERIMENTAL SETUP AND RESULTS

Given an IBVP (Equation 1), we define a compact set $K \subset \mathcal{X}$ of initial conditions and specify a sampling procedure. We parametrize the Hypernetwork $H_\Phi$ using a Multi-Layer Perceptron (MLP). The input to $H_\Phi$ is a batch of discretized initial conditions $u_0 \in K$. From the output, we compute the parameters of the Target Network $T_\theta$, itself an MLP with low-rank hidden weights. The input to $T_\theta$ is a batch of space-time coordinates $(t, x)$. As per Equation 3, the output of $T_\theta$ is then reparametrized to satisfy the initial condition. Using automatic differentiation, we compute the derivatives of the output of $T_\theta$ w.r.t. $x$ and $t$ and construct the residual loss $\mathcal{L}_{\text{PDE}}$. The boundary loss $\mathcal{L}_{\text{BC}}$ is computed in a supervised manner. Finally, we build the total loss $\mathcal{L}$ as a weighted sum of $\mathcal{L}_{\text{PDE}}$ and $\mathcal{L}_{\text{BC}}$ and differentiate the parameters $\Phi$ of $H_\Phi$ w.r.t. $\mathcal{L}$ to train the Hypernetwork using Adam (Kingma & Ba, 2015). Periodic loss weight calculations normalize the magnitudes of the updates caused by the different loss components. Section A.9 contains a full description of the algorithm.

We evaluate NPR on two one-dimensional example problems, the heat equation and Burgers equation. We perform ablations regarding the hidden dimension and the rank of the weights in the Target Network and compare to Physics-Informed DeepONets. Finally, we demonstrate the adaptability of our model by evaluating and fine-tuning on out-of-distribution initial conditions. All experiments are carried out on a single NVIDIA Tesla V100 GPU. The significantly higher training time for the heat equation is due to the presence of second order derivatives and the more involved sampling scheme.

**Heat equation.** We consider the heat equation with constant Dirichlet boundary conditions on $\Omega = [0, 1]$ and $T = 1$. The set $K$ is parametrized by Fourier polynomials with a fixed number of coefficients and bounded absolute value, see Section A.2.1 for details. We compute a reference solution using a finite difference scheme for twelve different initial conditions from the proposed distribution, and compute the $L^1$, $L^2$, and $L^\infty$ errors between reference solution and model output.

We fine-tune a trained model to the out-of-distribution initial condition $u_0(x) = 5x + 3\sin(4\pi x)$. To achieve this, we compute the output of the Hypernetwork $H_\Phi(u_0)$ and use it as parameters for Target Network $T_\theta$. We discard $H_\Phi$ and regard $T_\theta$ as a conventional PINN, training it for 200 steps. The results are shown in Figure 3. Without fine-tuning, the model performs poorly, but after only 200 iterations ($\approx 2$ seconds on a AMD Ryzen 7 PRO 6850U *CPU*) the model adapts to the new initial condition. See Section A.10 for details on fine-tuning.

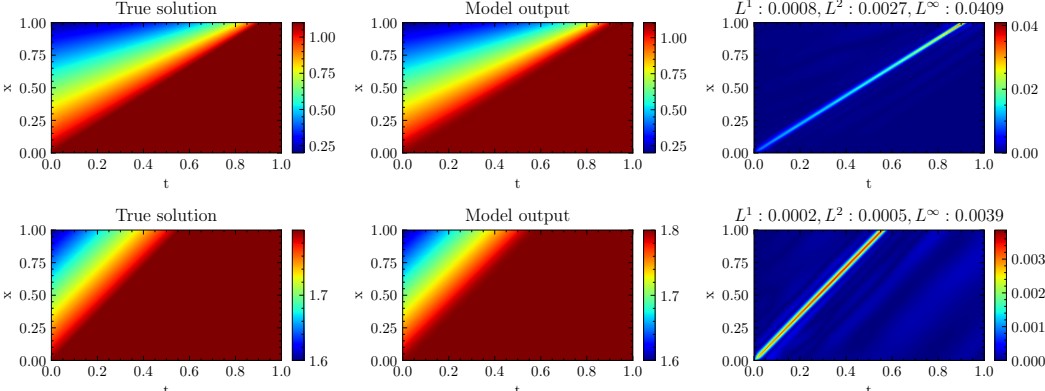

Figure 3: The heat equation with the out-of-distribution condition $u_0(x) = 5x + 3\sin(4\pi x)$. We plot the reference solution (left), the absolute difference to the reference solution before fine-tuning (middle) and after fine-tuning (right).

Figure 4: The Burgers equation: The first row shows the results for the initial condition $u_0(x) = -0.9x + 1.1$ and the second row for $u_0(x) = -0.2x + 1.8$.

**Burgers equation.** To showcase that our approach can also handle nonlinear dynamics, we consider the (inviscid) Burgers equation on $\Omega = [0, 1]$ and $T = 1$. We impose a constant Dirichlet boundary condition at $x = 0$ and parametrize the set of initial conditions $K$ by affine functions $u_0(x) = ax + b$ with $a \in [-1, 0]$ and $b \in [1, 2]$. In this setting we can guarantee that no shocks occur and we can give an explicit formula for the solution, see Section A.2.2 for details. Figure 4 shows how our model captures the induced nonlinear dynamics at different output scales. Close to the formation of a shock, the performance degrades slightly, which is expected given the increasingly steep nature of the solution in this region.

We present the results for both equations in Table 1. For the heat equation, the performance in terms of $L^1$ and $L^2$ errors is comparable to the DeepONet, the $L^\infty$ error however is much lower across all considered configurations. This is likely due to the enhanced expressivity of NPR, which enables it to precisely capture the solution in the whole space-time domain. For the Burgers equation, NPR outperforms the DeepONet in all metrics and configurations, even though we increased the sizes of the Branch Network and Trunk Network as compared to the heat equation, underlining the superiority of NPR in capturing nonlinear dynamics. By design, NPR requires more parameters than the DeepONet, as the the output dimension of the Hypernetwork is a compressed representation of the parameters of a full NN. Increasing the amount of DeepONet parameters to a comparable amount gave worse results than the ones reported here.

## 4    CONCLUSION

We have presented *Neural Parameter Regression*, merging Hypernetworks and PINNs for self-supervised learning of PDE solution operators. Utilizing low-rank matrices for compact parameterization and ensuring initial condition compliance, our approach efficiently learns PDE solution operators and adapts swiftly to new conditions, promising advancements in the field.

While our method shows promising results, it shares a fundamental shortcoming of the DeepONet, in the sense that it is only applicable to operators between functions spaces of relatively low di-

Table 1: Comparative results for the heat and Burgers equations. *# Target* and *# Hyper* denote the total number of parameters in the Target Network and the Hypernetwork, respectively. For the DeepONet, they refer to the number of parameters in the Trunk and Branch Networks.

| Equation | Metric | Hidden dim 32 | | | Hidden dim 64 | | | DeepONet |
|----------|--------|--------|--------|---------|--------|--------|---------|----------|
| | | Rank 4 | Rank 8 | Rank 16 | Rank 4 | Rank 8 | Rank 16 | |
| Heat | $L^1$ | 0.0037 | 0.0028 | 0.0037 | 0.0036 | 0.0026 | **0.0021** | 0.0026 |
| | $L^2$ | 0.0051 | 0.0037 | 0.0047 | 0.0046 | 0.0031 | **0.0030** | 0.0036 |
| | $L^\infty$ | 0.0311 | 0.0171 | 0.0165 | 0.0166 | 0.0268 | **0.0159** | 0.0349 |
| | *# Target* | 993 | 1761 | 3297 | 1985 | 3521 | 6593 | 4320 |
| | *# Hyper* | 79137 | 129057 | 228897 | 143617 | 243457 | 443137 | 16672 |
| | Training Time | 62 min | 65 min | 69 min | 67 min | 71 min | 76 min | 47 min |
| Burgers | $L^1$ | 0.0007 | 0.0006 | **0.0004** | 0.0005 | 0.0006 | 0.0005 | 0.0011 |
| | $L^2$ | 0.0023 | 0.0022 | **0.0014** | 0.0016 | 0.0019 | 0.0017 | 0.0030 |
| | $L^\infty$ | 0.0282 | 0.0276 | **0.0206** | 0.0218 | 0.0238 | 0.0223 | 0.0328 |
| | *# Target* | 993 | 1761 | 3297 | 1985 | 3521 | 6593 | 14752 |
| | *# Hyper* | 79137 | 129057 | 228897 | 143617 | 243457 | 443137 | 57888 |
| | Training Time | 16 min | 17 min | 20 min | 17 min | 20 min | 24 min | 14 min |

mensional domains. This limitation arises from the exponential increase in the number of sensors needed to meaningfully represent functions as the dimensionality of their domain grows. Further, the considered examples do not show challenging behaviour such as sharp local features. We leave the exploration of these limitations to future work.

ACKNOWLEDGMENTS

This research was partially funded by the Deutsche Forschungsgemeinschaft (DFG, German Research Foundation) under Germany's Excellence Strategy – The Berlin Mathematics Research Center MATH+ (EXC-2046/1, project ID: 390685689), as well as through the project A05 in the Sonderforschungsbereich/Transregio 154 Mathematical Modelling, Simulation and Optimization using the Example of Gas Networks.

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

# A  APPENDIX

## A.1  RELATED WORK

Our study extends the framework of PINNs and operator learning, a domain significantly advanced by the introduction of Physics-Informed DeepONets (Wang et al., 2021b). These models leverage automatic differentiation to learn mappings from initial conditions to solutions, demonstrating a significant extension of PINN methodologies. Subsequent enhancements to the DeepONet architecture, including a weighting scheme based on Neural Tangent Kernel (NTK) theory (Jacot et al., 2018), were introduced by Wang et al. (2022a). Additionally, Wang & Perdikaris (2023) proposed learning the solution operator for short intervals and iteratively applying it for long-term integration.

This paper seeks to extend the concept of PINNs into a more dynamic and versatile framework. Inspired by the recent advancements in Physics-informed DeepONets (Wang et al., 2021b), our approach focuses on explicitly learning solution operators of PDEs. By regressing the parameters of an NN to act as a PINN, we propose a method that not only adapts to varying initial and boundary conditions but also facilitates faster solution inference compared to traditional numerical methods. Outside of the field of Physics Informed Machine Learning, this concept is known as *Hypernetworks* (Ha et al., 2017). Chauhan et al. (2023) provide a comprehensive overview over existing hypernetwork approaches. de Avila Belbute-Peres et al. (2021) have already applied Hypernetworks in the context of PINNs, solving the parametric Burgers equation and the parametric Lorenz Attractor. We take inspiration from their method and apply it in the context of operator learning.

Parallel to these developments, Li et al. (2021) also explored a two-stage training process, similar to our approach. Theoretical underpinnings, such as bounds on the approximation error of PINNs, were explored by De Ryck & Mishra (2022). Innovations like finite basis PINNs (Moseley et al., 2023) and comprehensive surveys on Physics-Informed machine learning (Hao et al., 2023) further enrich the context of our work. Studies focusing on importance sampling (Daw et al., 2022) and extrapolation capabilities of DeepONets (Zhu et al., 2023) also provide valuable insights into the evolving landscape of PINN research.

Zanardi et al. (2023) have also applied HyperNetworks to Physics-Informed problems and also have used LoRa weight updates to a base model. We differ from this by inherently parametrizing the output net as a low-rank model. Cho et al. (2023) have also applied Hypernetworks to PINNs and parametrized the networks via low-rank matrices, but also in the setting of parametric PDEs with fixed initial conditions. To the best of our knowledge, this work is the first to extend the concept of HyperNetworks to the setting of learning solution operators of PDEs.

## A.2  PARTIAL DIFFERENTIAL EQUATIONS

Equation 1 describes a wide range of initial boundary value problems (IBVPs) for PDEs. The operator $\mathcal{B}$ can describe many different types of boundary conditions, e.g., Dirichlet, Neumann or Robin boundary conditions. We point out that the operator $\mathcal{N}$ may depend on $x$, $t$ and additional parameters $c$, i.e. $\mathcal{N} = \mathcal{N}(u, x, t, c)$, where we have omitted the dependence on $x$, $t$ and $c$ in the notation for the sake of readability. We only consider solution operators on compact subsets of $\mathcal{X}$, as compactness is an essential requirement for the universal approximation theorem for operators (Chen & Chen, 1995), on which the DeepONet framework was built.

A *solution* to Equation 1 in the classical sense is a function $u : [0, T] \times \Omega \rightarrow \mathbb{R}$ which satisfies the PDE and the initial- and boundary conditions. In this setting, the space $\mathcal{X}$ is usually chosen as $C^k(\Omega, \mathbb{R})$[1], where $k$ is high enough to allow for evaluations of the differential operator $\mathcal{N}$. The space $\mathcal{Y}$ is then usually chosen as $C([0, T] \times \Omega, \mathbb{R})$. Many other solution concepts exist, e.g. weak solutions, mild solutions or viscosity solutions. We refer to the classical literature on PDEs (Evans, 1998) for an overview of solution concepts.

---

[1]$C^k(\Omega, \mathbb{R})$ denotes the $k$-times continuously differentiable functions from $\Omega$ to $\mathbb{R}$.

### A.2.1 HEAT EQUATION

The one-dimensional heat equation with constant Dirichlet boundary conditions on $\Omega = [0,1]$ and $T = 1$, is described by

$$\partial_t u = \kappa \partial_{xx} u \text{ for } (t,x) \in [0,1] \times \Omega \tag{4a}$$

$$u(0,x) = u_0(x) \text{ for } x \in \Omega \tag{4b}$$

$$u(t,0) = u_0(0); \ u(t,1) = u_0(1) \text{ for } t \in [0,1]. \tag{4c}$$

The initial conditions $u_0$ are all assumed to be in a compact set $K$, which we parametrize in Fourier space. Concretely, we denote by $C_n^c$ for $n \in \mathbb{N}$ and $c > 0$ the set of functions $f : [0,1] \to \mathbb{R}$ of the form

$$f(x) = a_0 + \sum_{i=1}^{n} a_n \sin(2\pi n x) + b_n \cos(2\pi n x)$$

with

$$\max_{i=1,\ldots,n} \max\{|a_i|, |b_i|\} \leq c.$$

Here, $n$ controls the number of Fourier coefficients and $c > 0$ controls the maximal amplitudes. For fixed $n$ and $c$ the elements of $C_n^c$ are pointwise bounded and equicontinuous, hence by the Arzela-Ascoli theorem $C_n^c$ is a compact subset of $C(\Omega, \mathbb{R})$. This parametrization allows for a straightforward sampling procedure, by simply sampling the coefficients $a_i$ and $b_i$ uniformly from $[-c, c]$. Especially for larger values of $n$, it makes sense to sample coefficients from a smaller interval (e.g. $[-\frac{c}{n}, \frac{c}{n}]$), as otherwise the high frequency components will cause very high slopes in the sampled functions. While (for e.g., $n = 3$ and $c = 2$) our sampled functions look quite similar to the samples from a gaussian random field (GRF) in Wang et al. (2022a), we point out that our sampling procedure produces samples from a compact subset of $C(\Omega, \mathbb{R})$, while the GRF approach does not.

The hyperparameters for the sampling procedure are detailed in Table 2.

### A.2.2 BURGERS EQUATION

As a second example, we consider the one-dimensional (inviscid) Burgers equation on $\Omega = [0,1]$ and $T = 1$ with a constant Dirichlet boundary condition at $x = 0$, i.e.,

$$\partial_t u = -u \partial_x u \text{ for } (t,x) \in [0,1] \times \Omega \tag{5a}$$

$$u(0,x) = u_0(x) \text{ for } x \in \Omega \tag{5b}$$

$$u(t,0) = u_0(0) \text{ for } t \in [0,1] \tag{5c}$$

The initial conditions are parametrized as affine functions $u_0(x) = ax + b$ with $a \in [-1, 0]$ and $b \in [1, 2]$. It is well-known (Chandrasekhar, 1943) that the solution on the time interval $[0,1]$ in this case is given by

$$u(t,x) = \min\left\{ \frac{ax + b}{at + 1}, b \right\}. \tag{6}$$

After one time unit, or more precisely at $t = -\frac{1}{a}$, the solution develops a shock and constructing solutions is much more involved. We refer to Toro (2009) for details on how to treat shock waves in the context of hyperbolic PDEs theoretically as well as numerically.

### A.3 HYPERPARAMETERS

Table 2 details the hyperparameters used in the experiments. Two unconventional choices are the $\sin$ activation function and the use of the mean absolute error as a loss function. We also experimented with $\text{relu}$ (for the Hypernetwork) and $\tanh$ activations, however found that $\sin$ consistently performed best. We also ran experiments using the mean squared error as loss function, however found that the mean absolute error performed better. Following Zimmer et al. (2023b), we employ a linearly decaying learning rate schedule after a linear warmup for the first 10% of the training steps.

All experiments were carried out on a NVIDIA Tesla V100 GPU. The training time for the heat equation was between 60 and 75 minutes (depending on the hidden dimension and rank out the

Table 2: Hyperparameters in our experiments.

| Parameter | Heat equation | Burgers equation |
|---|---|---|
| Number of fourier coefficients $n$ | 3 | - |
| Maximal absolute amplitude $c$ | 2 | - |
| $a_{\text{low}}$ | - | -1 |
| $a_{\text{high}}$ | - | 0 |
| $b_{\text{low}}$ | - | 1 |
| $b_{\text{high}}$ | - | 2 |
| Number optimization steps | $2 \times 10^{16} = 65536$ | $2 \times 10^{16} = 65536$ |
| Collocation batch size $n_{\text{pde}}$ | 2048 | 2048 |
| Learning rate | $1 \times 10^{-3}$ | $1 \times 10^{-3}$ |
| Optimizer | Adam | Adam |
| Loss function | Mean absolute error | Mean absolute error |
| Number of hidden layers in $H_\Phi$ | 4 | 4 |
| Hidden dimension in $H_\Phi$ | 64 | 64 |
| Activation in $H_\Phi$ | sin | sin |
| Number of hidden layers in $T_\theta$ | 4 | 4 |
| Activation in $T_\theta$ | sin | sin |
| Hidden dimension in $T_\theta$ | variable, see Table 1 | variable, see Table 1 |
| Output rank in $T_\theta$ | variable, see Table 1 | variable, see Table 1 |

target network) and between 15 and 22 minutes for the Burgers equation. The difference is due to higher order derivatives in the heat equation.

For the DeepONets in Table 1 we used a Branch Network with four hidden layers and a hidden dimension of 64 and a Trunk Network with four layers and a hidden dimension of 32. Across the considered values for these hyperparameters, this configuration gave the best result. For the Burgers equation, we used a Branch Network with a hidden dimension of 128 and a Trunk Network with a hidden dimension of 64. Still, the results were not competitive with NPR.

### A.3.1 EVALUATION DETAILS

To calculate the metrics for the heat equation, we compute a reference solution on a grid of size $500^2$ using a finite difference scheme. Concretely, we discretize the domain $[0, 1] \times [0, 1]$ by using equidistant grid points $x_1, \ldots, x_n$ and $t_1, \ldots, t_m$ with $m = n = 500$.

We then compute the absolute value of the difference $d_{\text{abs},ij}$ between the reference solution and the model output on this grid and the errors as

$$L^1 = 500^{-2} \sum_{x_i,t_j} d_{\text{abs},ij}, \quad L^2 = 500^{-2} \sqrt{\sum_{x_i,t_j} d_{\text{abs},ij}^2}, \quad L^\infty = \max_{x_i,t_j} d_{\text{abs},ij}.$$

For the Burgers equation, we use the explicit formula from Equation 6 and compute the errors in the same way as for the heat equation.

### A.4 PHYSICS-INFORMED NEURAL NETWORKS

The idea of PINNs (Raissi et al., 2019) is to design a loss function, which incorporates all the components of equation 1 and is zero if and only if the network solves the PDE. The typical approach is to parametrize the solution by an NN $u_\theta$ with parameters $\theta$ and introduce three loss components:

$$\mathcal{L}_{\text{PDE}} = \int_{[0,T]\times\Omega} \|\partial_t u_\theta(t, x) - \mathcal{N}(u_\theta(t, x))\| \, \mathrm{d}x\mathrm{d}t$$

$$\mathcal{L}_{\text{IC}} = \int_\Omega \|u_\theta(x, 0) - u_0(x)\| \, \mathrm{d}x$$

$$\mathcal{L}_{\text{BC}} = \int_{[0,T]\times\partial\Omega} \|\mathcal{B}(u_\theta(t, x))\| \, \mathrm{d}x\mathrm{d}t.$$

The partial derivatives needed to evaluate $\partial_t u_\theta$ and $\mathcal{N}u$ are computed using automatic differentiation and the norm $\|\cdot\|$ appearing in the integrals is typically the $L^2$ norm. The total loss is given by $\mathcal{L} = \lambda_{\text{PDE}}\mathcal{L}_{\text{PDE}} + \lambda_{\text{IC}}\mathcal{L}_{\text{IC}} + \lambda_{\text{BC}}\mathcal{L}_{\text{BC}}$, where $\lambda_{\text{PDE}}$, $\lambda_{\text{IC}}$ and $\lambda_{\text{BC}}$ are loss weights which can be chosen as hyperparameters or adapted dynamically (Wang et al., 2021a; 2022b).

## A.5 (Physics-Informed) DeepONets

The DeepONet (Lu et al., 2021a) architecture gives an approximation of any operator $G$ implicitly, that means: Given a function $u$ and a value $y = (t, x)$, the NN returns an approximation of $G(u)(y)$. While this approach has shown to be successful in many applications, it has some drawbacks. First, the mapping $u \mapsto G(u)$ is a bit hidden in the architecture. It can be recovered by evaluating the Branch Network $G_{\text{branch}}$ at $u$ and then considering the mapping $(t, x) \mapsto G_{\text{trunk}}(t, x) \cdot G_{\text{branch}}(u)$. This type of recovery can be useful if one wants to evaluate the function $G_{\text{branch}}(u)$ at many points without having to run the full network. This however illustrates another shortcoming of the Deep-ONet architecture: An unneccesarily large amount of the complexity needs to be allocated at the Trunk Network, which must produce rich enough representations of the inputs $(t, x)$ to be able to approximate $G(u)(t, x)$ for *any* $u$ by an operation as simple as the dot product, *without* knowledge of $u$. This issue has been addressed in Wang et al. (2022a) by introducing additional encoders which mix the hidden representations of the Branch Network and Trunk Network at every step. However, introducing these encoders makes the recovery of the mapping $u \mapsto G(u)$ impossible, as now for each pair $(u, (t, x))$ the whole architecture needs to be evaluated.

Building upon the PINN framework, the Physics-informed DeepONet (Wang et al., 2021b) applies these principles in an operator learning context. It harnesses the power of automatic differentiation and extends the capabilities of PINNs to learn mappings between function spaces while respecting physical constraints. It is worth emphasizing that both approaches do not require any labeled training data (except for initial- and boundary conditions, which are needed to make the problem well-posed) and can thus be considered as self-supervised learning methods. This is particularly crucial in the context of learning solution operators, as obtaining training data involves applying numerical solvers. Recently, Hasani & Ward (2024) have proposed to generate training data for neural operators which avoids solving the PDE numerically.

## A.6 Physics-constrained Neural Networks

In the specific setting of learning the mapping from an initial condition to the solution at a given time, a model $N_\theta$ is posed with the challenge of learning an identity mapping for $t = 0$, i.e. $N_\theta(u_0, 0, x) = u_0(x)$ must hold for all $x \in \Omega$. Lu et al. (2021b) proposed to parametrize the output in a way that the initial condition is always satisfied. Concretely, given any model $N_\theta$, a new model $\hat{N}_\theta$ is defined as follows:

$$\hat{N}_\theta(u_0, t, x) = (1 - \alpha(t))N_\theta(u_0, t, x) + \alpha(t)u_0(x).$$

Here, $\alpha : [0, T] \to [0, 1]$ is a function which is 1 for $t = 0$ and 0 for $t = T$, e.g $\alpha(t) = 1 - \frac{t}{T}$.[2] This was adopted by Brecht et al. (2023) in the context of operator learning. This way, we do not only mitigate the problem of learning the identity, but also eliminate the need for the $\mathcal{L}_{\text{IC}}$ term in the loss function, thus reducing the number of hyperparameters and making the problem easier. In the case of Dirichlet boundary conditions, we can apply the same procedure to those by parametrizing the output by a function $\beta : [0, T] \times \partial\Omega \to \mathbb{R}^d$ and defining the new model as:

$$\hat{N}_\theta(u_0, t, x) = (1 - \beta(x))N_\theta(u_0, t, x) + \beta(x)u_b(t), \tag{7}$$

where $u_b : [0, T] \to \partial\Omega$ describes the (non-homogenuous) Dirichlet boundary condition and $\beta : \partial\Omega \to [0, 1]$ is a function which is 1 for $x \in \partial\Omega$ and between 0 and 1 for $x \in \Omega$. This eliminates $\mathcal{L}_{\text{BC}}$ from the loss function. We point out that neither a closed form of $u_0$ nor $u_b$ is required. If we only have samples of $u_0$ and $u_b$ at certain points, we can simply interpolate them (e.g. using linear interpolation or splines).

---

[2]The factor $\alpha(t)$ in front of the $u_0$ term is not necessary to achieve the hardcoding, however we found that it works better.

While we found that hardcoding the boundary conditions increases the speed at which our models converge, it is not strictly necessary. Hardcoding the initial conditions however is crucial. This is interesting, as it hints at the following: Regressing the parameters of an NN to act as a PINN is a very difficult task, but clearly it will be even harder if the networks are supposed to learn complicated mappings. Thus, our study demonstrates that hardcoding initial conditions greatly simplifies the learning problem and should be considered in any application of PINNs.

## A.7 HYPERNETWORKS

Ha et al. (2017) have introduced the approach of one NN learning the parameters of another NN and have coined it *Hypernetworks*. We refer to Chauhan et al. (2023) for a recent review on the topic.

## A.8 LOW-RANK PARAMETRIZATION OF MLPS

A multilayer perceptron (MLP) is a NN with an input layer, an output layer and one or more hidden layers. Each layer consists of a linear transformation followed by a non-linear activation function. The linear transformation is parametrized by a matrix $W$ and a bias vector $b$. Concretely, for an MLP with $n$ hidden layers and hidden dimension $d$, the output is defined as:

$$\begin{aligned}
\mathrm{MLP}(x) &= W_{n-1}h_{n-1} + b_{n-1} \\
h_i &= \sigma(W_i h_{i-1} + b_i) \text{ for } i = 2, \ldots, n-1 \\
h_1 &= \sigma(W_1 x + b_1)
\end{aligned}$$

where $x$ is the input, $h_i$ is the output of the $i$-th layer, $W_i$ are the weight matrices and $b_i$ are the bias vectors. $\sigma$ is a nonlinear function, often called the *activation* and is typically chosen as $\tanh$ or ReLU.

If we choose to parametrize the Target Network by a multilayer perceptron (MLP) with $n_{\text{hidden}}$ hidden layers, each with a hidden dimension of $d_{\text{hidden}}$, the number of parameters is given by

$$d_p = d_{\text{input}}d_{\text{hidden}} + d_{\text{hidden}} + (n_{\text{hidden}} - 1)(d_{\text{hidden}}^2 + d_{\text{hidden}}) + d_{\text{hidden}}d_{\text{output}} + d_{\text{output}}. \tag{9}$$

Since this scales quadratically in the hidden dimension $d_{\text{hidden}}$, it quickly becomes intractable as it blows up the final layer of our network. Therefore, motivated by the success of LoRA (Hu et al., 2022; Zimmer et al., 2023a), we parametrize the weights of the hidden layers as the product of 2 matrices, i.e. $W_i = A_i B_i$ for $i = 1, \ldots, n_{\text{hidden}} - 1$ with $A_i, B_i^T \in \mathbb{R}^{d_{\text{hidden}} \times r}$, where $r \ll d_{\text{hidden}}$ is the rank of the matrices and therefore the rank of the matrix $W_i$ is also at most $r$. This reduces the number of parameters to

$$d_p = d_{\text{input}}d_{\text{hidden}} + d_{\text{hidden}} + 2(n_{\text{hidden}} - 1)(r d_{\text{hidden}} + d_{\text{hidden}}) + d_{\text{hidden}}d_{\text{output}} + d_{\text{output}} \tag{10}$$

Remarkably, for some PDEs we achieve good results with a rank as low as four, showing that by this approach we can learn approximations of the solution operator efficiently.

## A.9 DETAILS ON THE TRAINING ALGORITHM

**Hypernetwork and Target Network Architectures** Our Hypernetwork $H_\Phi$ is parametrized as a multilayer perceptron (MLP). The choice of the number of hidden layers, their dimensions, and the activation function ($\sin$ instead of the common $\tanh$) is made to allow for richer implicit neural representations, as suggested by Sitzmann et al. (2020). The Target network $T_\theta$ also follows an MLP architecture with low-rank hidden weights, designed to efficiently capture the dynamics of the problem.

**Sampling and Discretization of Initial Conditions** Initial conditions $u_0$ are sampled from a compact set $K \subset \mathcal{X}$ using a specified sampling procedure. The discretization of $u_0$ into input vectors or tensors for $H_\Phi$ in this work simply means taking equidistant samples of $u_0$ on the domain $\Omega$. The number of samples is a hyperparameter of the model ans was chosen to be 32 in this study.

**Periodic Update of Loss Weights** The loss weights $\lambda_{\text{PDE}}, \lambda_{\text{IC}}, \lambda_{\text{BC}}$ are updated periodically to normalize the magnitudes of the gradient updates. Considering the case that all loss components are present (neither initial- nor boundary conditions are hardcoded), every 100 steps we compute the loss weights as follows:

1. Sample a batch for each of the loss components.

2. Compute $\left\| \frac{\partial \Phi}{\partial \mathcal{L}_i} \right\|$ for $i$ in $\{\text{PDE}, \text{IC}, \text{BC}\}$ using automatic differentiation.

3. Compute the total magnitude of the updates $M = \sum_i \left\| \frac{\partial \Phi}{\partial \mathcal{L}_i} \right\|$.

4. Set $\lambda_i = M \left\| \frac{\partial \Phi}{\partial \mathcal{L}_i} \right\|^{-1}$ for $i$ in $\{\text{PDE}, \text{IC}, \text{BC}\}$.

**Minibatch Construction and Optimization** We explain in more detail how minibatches are constructed for computing the loss components. For $\mathcal{L}_{\text{PDE}}$, we sample $n_{\text{pde}}$ initial conditions from $K$ as well as $n_{\text{pde}}$ space-time coordinates $(t, x)$ from $[0, T] \times \Omega$. The initial conditions $u_0$ are discretized into vectors $\hat{u}$, so a batch which is input to the hypernetwork is of the form $\hat{u}_i, (t_i, x_i)$, where $i$ is the index for the position in the minibatch. After passing this batch through $H_\Phi$, we obtain a batch of vectors $\hat{v}_i$, which we reshape into a batch of NNs $T_{\theta,i}$. We then evaluate (in parallel) the NNs $T_{\theta,i}$ at the points $(t_i, x_i)$ and use automatic differentiation to compute the derivatives $\partial_t T_{\theta,i}$ and $\mathcal{N} T_{\theta,i}$. Finally, we compute the loss component $\mathcal{L}_{\text{PDE}}$ and the gradient of the hypernetwork parameters $\Phi$ w.r.t. $\mathcal{L}_{\text{PDE}}$ using automatic differentiation. For $\mathcal{L}_{\text{BC}}$, we sample $n_{\text{bc}}$ initial conditions from $K$ as well as $n_{\text{bc}}$ space-time coordinates from $\partial\Omega \times [0, T]$. Again, the initial conditions are discretized and passed through the hypernetwork, after which we evaluate the NNs at the sampled points and compute the loss component $\mathcal{L}_{\text{BC}}$ and its gradient in a supervised manner. For $\mathcal{L}_{\text{IC}}$, the batch consists of $n_{\text{ic}}$ initial conditions sampled from $K$ together with points $x_i$ sampled from $\Omega$. We pass the discretized initial conditions through the hypernetwork and evaluate the NNs at $(0, x_i)$. We then compute the loss component $\mathcal{L}_{\text{IC}}$ and its gradient in a supervised manner. Finally, we construct the total loss $\mathcal{L} = \lambda_{\text{PDE}} \mathcal{L}_{\text{PDE}} + \lambda_{\text{IC}} \mathcal{L}_{\text{IC}} + \lambda_{\text{BC}} \mathcal{L}_{\text{BC}}$ and compute the gradient of the Hypernetwork parameters $\Phi$ w.r.t. $\mathcal{L}$ using automatic differentiation. We then update the Hypernetwork parameters using the Adam Algorithm. We present the full training algorithm in Algorithm 1.

## A.10 FINE-TUNING PROCEDURE

To enhance the adaptability of our model for both challenging in-distribution initial conditions and out-of-distribution examples, we propose a systematic procedure. Our model, by design, outputs NN parameters corresponding to each given initial condition. This capability enables us to input any initial condition, within the constraints of our sensor resolution[3], to obtain a specific set of NN parameters. To increase expressivity, we then "unfold" this network by explicitly computing the low-rank product matrices $A_i B_i$ for the hidden weights. Subsequently, the Branch Network is omitted, treating the resulting network as a conventional PINN. Remarkably, this adaptation process, which requires only a few hundred steps or a matter of seconds, significantly reduces the error for a wide range of initial conditions, including out-of-distribution as shown in Figure 3.

---

[3]This refers to initial conditions that can be meaningfully represented given the chosen number of sensors.

---

**Algorithm 1** The full training algorithm for Neural Parameter Regression

---

**Input:** Initial condition distribution, domain $\Omega \times [0, T]$
**Hyperparameters:**
$N_{\text{steps}}$ (Total training steps)
$B_{\text{collocation}}$ (Collocation batch size)
$B_{\text{IC}}$ (Initial condition batch size)
$B_{\text{BC}}$ (Boundary condition batch size)
$\omega_{\text{freq}}$ (Loss weight update frequency)
**Output:** Trained Hypernetwork parameters
Initialize Hypernetwork parameters
**for** $i = 1$ to $N_{\text{steps}}$ **do**
    **if** $i \bmod \omega_{\text{freq}}$ is 0 **then**
        Update loss weights $\lambda_{\text{PDE}}, \lambda_{\text{IC}}, \lambda_{\text{BC}}$
    **end if**
    Sample batch of $B_{\text{collocation}}$ initial conditions $u_0$
    Discretize $u_0$ into vectors or tensors
    Sample collocation points $(t, x)$ from $\Omega \times [0, T]$
    Evaluate and reshape Hypernetwork output to get NNs
    Compute the differential operator $\mathcal{N}$ and $\partial_t$ using automatic differentiation
    Compute $\mathcal{L}_{\text{PDE}}$
    **if** initial condition not hardcoded **then**
        Sample batch of $B_{\text{IC}}$ initial conditions $u_0$ for $\mathcal{L}_{\text{IC}}$
        Compute $\mathcal{L}_{\text{IC}}$
    **end if**
    **if** boundary condition not hardcoded **then**
        Sample batch of $B_{\text{BC}}$ boundary conditions $u_0$ for $\mathcal{L}_{\text{BC}}$
        Compute $\mathcal{L}_{\text{BC}}$
    **end if**
    Construct total loss $\mathcal{L} = \lambda_{\text{PDE}}\mathcal{L}_{\text{PDE}} + \lambda_{\text{IC}}\mathcal{L}_{\text{IC}} + \lambda_{\text{BC}}\mathcal{L}_{\text{BC}}$ and compute the gradient of the Hypernetwork's parameters $\Phi$ w.r.t. Ł using automatic differentiation
    Update Hypernetwork parameters using a variant of stochastic gradient descent (e.g. Adam)
**end for**

---

