# OpenReview forum: "Neural Parameter Regression for Explicit Representations of PDE Solution Operators"
_ICLR.cc/2024/Workshop/AI4DiffEqtnsInSci — AI4DiffEqtnsInSci @ ICLR 2024 Poster_

### Official Review · Reviewer_YoR2 · 2024-02-15
**Introduces Neural Parameter Regression (a model with two networks) for operator learning**

**Rating:** 4
**Confidence:** 5

**Review:**

The paper introduces Neural Parameter Regression or NPR for operator learning. The idea of the paper is to encode the imposed initial condition into a latent space which is then mapped to the solution space via a secondary neural network. The latter NN additionally takes (x, t) as inputs and avoids trivial solutions by reparametrizing the output such that the model learns the deviation from the initial conditions.

I believe the paper has a novel idea but it is not analyzed with sufficient depth. I can see the following main issues with the current state of their approach: (1) in high spatial dimensions, how would you handle the large encoding space (i.e., the method does not seem to scale well to cases where the initial condition is specified in 3D), (2) any version of the model considered in Table 1 has much more parameters than DeepONet so I am not confused about its higher performance (i.e., a fair comparison would assign roughly the same number of parameters to each model), (3) is the method able to handle local features (e.g., high frequency or large gradients)? The examples used are very simple. (4) the method is discretization dependents (due to discretization of u0) so perhaps you can use PCA to remove this limitation.

---

### Official Review · Reviewer_5hJ1 · 2024-02-27
**Interesting idea with good experimental evaluation.**

**Rating:** 6
**Confidence:** 4

**Review:**

The authors present a novel approach to operator learning with the application of learing solution operators for PDEs in mind. The proposed method applies the concept of Hypernetworks to the realm of Scientific Machine Learning and operator learning in particular.

The idea is simple and presented in a straight-forwards fashion. The idea is tested on two different PDE problems (Burgers equation and Heat equation) and hyperparameters are disclosed to make the setup reproducible. My main criticism is the limited evaluation of the method - as these two problems are fairly simple toy problems. As the self-supervised nature of physics-informed losses do not require extensive generation of training datasets, i would have welcomed more experiments, particularly in 2 spatial dimensions. Moreover, only DeepONets are considered as a baseline here. I understand that this is due to the problem setting. However, I would have welcomed a comparison to  other baselines such as Physics Informed Neural Operators (PINO).

Given that this is a workshop, I propose to accept the submission and encourage the authors to address my concern above, should they consider a full submission somewhere else.

---

### Meta-Review · Area_Chair_gNjq · 2024-03-01

**Recommendation:** Accept (Poster)

**Metareview:**

This author presents a novel method for operator learning. Despite there are some concerns from reviewer YoR2, regarding the experiment results and whether this method can handle local features, those concerns can be address through rewriting. I strongly recommend the authors to address both reviewers' concerns in the camera-ready version.

---

### Decision · Program_Chairs · 2024-03-01

Accept (Poster)